# The Association between Ambient PM_2.5_ and Low Birth Weight in California

**DOI:** 10.3390/ijerph192013554

**Published:** 2022-10-19

**Authors:** Jasmine Lee, Sadie Costello, John R. Balmes, Stephanie M. Holm

**Affiliations:** 1Division of Epidemiology, School of Public Health, University of California, Berkeley, CA 94704, USA; 2Division of Environmental Health Sciences, School of Public Health, University of California, Berkeley, CA 94704, USA; 3Division of Occupational and Environmental Medicine, Department of Medicine, University of California, San Francisco, CA 94143, USA

**Keywords:** air pollution, low birth weight, epidemiology, ecologic study

## Abstract

Previous studies have shown associations between air pollutants and low birth weight. However, few studies assess whether poverty and race/ethnicity are effect modifiers for this relationship. We used publicly available data on 7785 California census tracts from the California Communities Environmental Health Screening Tool (CalEnviroScreen). Multivariable linear regression was used to examine the association between outdoor PM_2.5_ and low birth weight (LBW), including stratification by poverty and race/ethnicity (as a proxy for experienced racism). A 1 µg m^−^^3^ increase in PM_2.5_ was associated with a 0.03% (95% CI: 0.01, 0.04) increase in the percentage of LBW infants in a census tract. The association between PM_2.5_ and LBW was stronger in census tracts with the majority living in poverty (0.06% increase; 95% CI: 0.03, 0.08) compared to those with fewer people living in poverty (0.02% increase; 95% CI: 0.00, 0.03). Our results show that exposure to outdoor PM_2.5_ is associated with a small increase in the percentage of LBW infants in a census tract, with a further increase in tracts with high poverty. The results for effect modification by race/ethnicity were less conclusive.

## 1. Introduction

In 2019, approximately 7% of live births in California were classified as low birth weight (less than 2500 g or ~5.5 pounds) [1]. Low birth weight (LBW) is an important predictor of infant mortality, especially during the neonatal period, and is also associated with increased risk of chronic conditions later in life, such as cardiovascular disease, obesity, and cognitive development [2]. There has not been a substantial decline in the prevalence of LBW infants in the U.S. during the past two decades [3]. Addressing possible risk factors for LBW may provide an opportunity to decrease the proportion of LBW infants and reduce the associated negative health outcomes.

Particulate matter less than 2.5 microns in diameter (PM_2.5_) is an air pollutant that is associated with adverse birth outcomes, including LBW, preterm birth, and still birth [4]. Multiple studies have examined the relationship between PM_2.5_ and LBW, and the majority of the epidemiological evidence shows that prenatal exposure to PM_2.5_ is associated with an increased risk of this outcome [5,6,7,8]. In a study conducted in China of over a million parent and child pairs the authors found that a 10 µg m^−^^3^ increase in PM_2.5_ was associated with a 20%, 18%, and 20% increase in the hazard of LBW for trimester 1, 2, and 3, respectively [5]. Another study in California found that exposure to PM_2.5_ slightly increased the odds of low birth weight where PM_2.5_ was assessed by distance from stationary monitors used for compliance with U.S. Environmental Protection Agency (EPA) air quality standards [6]. Recent systematic reviews and meta-analyses have also looked at the relationship between PM_2.5_ and LBW. One systematic review found that birth weight decreased by 27.55 g for every 10 µg m^−3^ increase in PM_2.5_ exposure [7]. Another systematic review and meta-analysis found that exposure to less than 10 µg m^−3^ of PM_2.5_ was associated with 15.58 g decrease in term birth weight while exposure to greater than 10 µg m^−3^ of PM_2.5_ was associated with 16.58 g decrease in term birth weight [8].

Socioeconomic status and experiencing discrimination are important determinants of health. Low-income communities, including many communities of color, are exposed to higher levels of air pollution and have higher proportions of infants with LBW [9,10]. At all income levels, people of color are disproportionately exposed to air pollutants [11]. Overall, people of color are more likely to live near sources of pollution, such as freeways and industrial facilities, compared to their white counterparts [12]. Few studies have assessed whether there is effect measure modification by income status or race/ethnicity (as a proxy for experienced structural and interpersonal racism) on the relationship between PM_2.5_ exposure and LBW [13]. Understanding the impacts of poverty and race/ethnicity on the association between PM_2.5_ and LBW may help policymakers in California target air pollution regulation to protect the most vulnerable communities and address environmental inequities.

The California Communities Environmental Health Screening Tool (CalEnviroScreen) is created by the California Office of Environmental Health Hazard Assessment (OEHHA). Using data from federal and state government sources, this tool identifies and maps communities in California that are most vulnerable to pollution. It contains data on various sources of pollution, health outcomes, and sociodemographic measures at the census tract level and has been updated 4 times to reflect secular trends [14]. Government agencies, such as the California Environmental Protection Agency (CalEPA) use the data from CalEnviroScreen to inform policy and allocate resources to disadvantaged communities [15].

In this study, we examine the relationship between ambient PM_2.5_ and LBW across the state of California at the census tract level, as well as investigate whether the relationship is modified by neighborhood-level poverty or percentage of people in different racial/ethnic groups (White, Hispanic, African American, and Asian American). We hypothesized that there would be a greater effect of PM_2.5_ on the percentage of infants with LBW among census tracts with higher proportions of households in poverty and for census tracts with higher percentages of underrepresented minorities compared to census tracts with lower percentages of either factor.

## 2. Materials and Methods

### 2.1. Study Sample

We used data from OEHHA’s CalEnviroScreen 3.0 and CalEnviroScreen 4.0 for census tracts within California. Compared to CalEnviroScreen 3.0, CalenviroScreen 4.0 contains more recent data for all indicators and improvements in the methods for measuring certain variables. We used both CalEnviroScreen 3.0 and CalEnviroScreen 4.0 because the data for the outcome (LBW) in CalEnviroScreen 4.0 were collected from 2009 to 2015, which most closely corresponds to the data for the exposure (PM_2.5_) in CalEnviroScreen 3.0 (which were collected from 2012 to 2014). In contrast, the LBW data in CalEnviroScreen 3.0 are from a time period before the exposure data. CalEnviroScreen contains information on 8035 census tracts in California. We excluded 250 census tracts (3.1%) that did not have data on PM_2.5_, percentage of low birth weight infants, or percentage of the population living in poverty. The final analytic sample contained 7785 census tracts (Figure 1).

### 2.2. PM_2.5_ Measurement

The annual mean concentration of PM_2.5_ collected from CalEnviroScreen 3.0, which was based on monitoring data from the California Air Resources Board’s (CARB) air monitoring network. To obtain measurements for each census tract, ordinary kriging was used to estimate the mean concentration at the center of the census tract. Quarterly estimates were averaged to obtain the annual mean and the annual means from 2012 to 2014 were averaged to obtain a single estimate of PM_2.5_ exposure for the census tract. For census tracts that did not have air monitors within 50 km, PM_2.5_ concentrations were obtained from satellite observations from the years 2006 to 2012 and averaged over that period to obtain the single estimate of PM_2.5_ exposure for the census tract. PM_2.5_ was measured in micrograms per cubic meter (µg m^−3^).

### 2.3. Low Birth Weight

The percent of LBW infants averaged over the years 2009 to 2015 for the 7785 census tracts were collected from CalEnviroScreen 4.0. Data on the percentage of low birth weight infants in a census tract were obtained from CalEnviroScreen 4.0 to ensure that most of the outcomes occurred concurrent with or after the reported PM_2.5_ data. OEHHA obtained data on births from the California Department of Public Health and geocoded them based on the mother’s residential address at the time of the birth.

### 2.4. Covariates

Covariates that could potentially confound the association between ambient PM_2.5_ level and LBW were identified using a directed acyclic graph. Covariates included percentage of households in poverty, percentage of race/ethnicity, and toxic releases from nearby facilities. All data for the covariates, except for the percentage of race/ethnicity, were from CalEnviroScreen 3.0 because race/ethnicity data were not available in CalEnviroScreen 3.0. Percentage of households in poverty was used as defined by CalEnviroScreen (the five-year estimate from the years 2011 to 2015 of the percentage of people living below twice the federal poverty level). Data on the percentage of race/ethnicity in a census tract were from CalEnviroScreen 4.0. Data on toxic releases from facilities were an average of toxicity-weighted concentrations of modeled chemical releases to air from facility emissions and off-site incineration from 2011 to 2013 in the census tract.

### 2.5. Statistical Analysis

We mapped the spatial distribution of PM_2.5_, LBW, and percentage of the population in poverty in California using QGIS Version 3.16.16.

We used multivariable linear regression to analyze the relationship between PM_2.5_ and LBW adjusted for percentage of the population in poverty, percentage of each racial/ethnic group, excluding white, and toxic releases from facilities in the census tract for the relationship between PM_2.5_ and LBW.

To examine poverty as an effect modifier, we used two times the federal poverty level as our threshold and stratified the data into census tracts with <50% of the population living below this level and census tracts with ≥50% of the population living below this level.

To look at race/ethnicity as an effect measure modifier, we created binary variables for the four largest race/ethnicity groups (Hispanic, White, African American, and Asian American) in California, identifying census tracts that were above the median for each race/ethnicity category. We conducted four stratified analyses in which we stratified the data into census tracts that were above and below or equal to the median for each of the four largest race/ethnicity groups. All stratified models were adjusted for the other race/ethnicity groups being above the median, the percentage of households in poverty, and toxic releases from facilities.

All analyses were conducted in R Version 4.1.1.

## 3. Results

Table 1 shows the characteristics of the 7785 census tracts with complete data used in the analysis. The average annual mean concentration of PM_2.5_ over three years across the census tracts was 10.4 micrograms per cubic meter (µg m^−3^), which is less than the current annual standard set by the U.S. EPA of 12 µg m^−3^ but higher than the World Health Organization’s annual guideline of 5 µg m^−3^ [16,17]. Figure 2 shows the maps of the spatial distribution of the salient variables.

PM_2.5_ was positively associated with low birth weight (LBW) in both adjusted and unadjusted models. After adjusting for percentage of the population living in poverty, percentage race/ethnicity, and toxic releases from facilities, the association between PM_2.5_ and LBW remained significantly positive. There was a 0.03% (95% CI: 0.01, 0.04) increase in the percentage of LBW infants for every 1 µg m^−3^ increase in average annual mean concentration of PM_2.5_ in a census tract after adjustment (Table 2).

In the stratified analysis, the percentage of people living in poverty was an effect modifier for the relationship between PM_2.5_ and LBW after adjustment. For census tracts with ≥50% of the population living in poverty, there was a 0.06% (95% CI: 0.03, 0.08) increase in the percentage of LBW infants for every 1 µg m^−3^ increase in average annual mean concentration of PM_2.5_, while for census tracts with <50% of the population living in poverty was associated with a 0.02% (95% CI: 0.00, 0.03) increase in the percentage of LBW infants after adjustment (Table 2). Thus, there was a larger association between PM_2.5_ and LBW in lower income areas compared to higher income areas. The association between PM_2.5_ and LBW was not statistically significant for census tracts with <50% of people living in poverty but for census tracts with ≥50% of people living in poverty, there was a statistically significant positive relationship between PM_2.5_ and LBW.

Figure 3 shows the results of our analysis when we looked at the race/ethnicity groups as effect modifiers after adjusting for percentage of the population living in poverty, the median percentage of the other race/ethnicity groups, and toxic releases from facilities. Among census tracts with below the median percentage of Hispanics, LBW increased by 0.04% (95% CI: 0.02, 0.06) for a 1 µg m^−3^ increase in PM_2.5_ while census tracts with greater than the median percentage increased by 0.02% (95% CI: 0.00, 0.04). Comparing census tracts that are above the median for the percent White population to those below the median, the tracts with a larger White population were associated with a 0.04% (95% CI: 0.02, 0.06) increase in LBW per 1 µg m^−3^ change in PM_2.5_ while tracts with a smaller White population were associated with a 0.01% (95% CI: −0.01, 0.03) increase in LBW. There was a 0.03% (95% CI: 0.02, 0.05) increase in the percentage of LBW infants for every 1 µg m^−3^ increase in the average annual mean concentration of PM_2.5_ in a census tract with below the median percentage of African Americans and a 0.01% (95% CI: −0.01, 0.03) increase in the percentage of LBW infants for census tracts with above the median percentage. Lastly, for census tracts with below the median percentage of Asian Americans, for every 1 µg m^−3^ increase in PM_2.5_, there was a 0.02% (95% CI: 0.00, 0.03) increase in the percentage of LBW infants while for census tracts with below the median, there was a 0.03% (95% CI: 0.01, 0.06) increase. In this analysis, the percentage of Hispanics, Whites, and African Americans in a census tract may be potential effect modifiers.

## 4. Discussion

This study aimed to understand the relationship between PM_2.5_ and low birth weight (LBW) in California and investigate whether poverty and race/ethnicity (as a proxy for experienced structural and interpersonal racism) modified that relationship. Although multiple studies have examined the relationship between PM_2.5_ and low birth weight, few studies have been in a statewide representative sample or assessed for effect modification by poverty or race/ethnicity on that relationship [8]. We found a small positive association between PM_2.5_ and LBW which is consistent with the results from previous studies [5,6,7,8]. A recent meta-analysis of the effect of maternal exposure to air pollution on the risk of low birth weight found that the risk of infants being born with LBW was 1.08 (95% CI: 1.04, 1.12) times greater for mothers exposed to ambient PM_2.5_ in the U.S. [18]. Our study adds to the existing body of literature by showing that the relationship between PM_2.5_ and LBW was modified by poverty but not by race/ethnicity at the census tract level.

The results for effect modification by race/ethnicity in this ecological analysis were surprising, given the results of prior studies showing communities of color having higher exposure to air pollution and being at increased risk of LBW [9,10]. We expected a stronger association between PM_2.5_ and LBW for census tracts with a high percentage of Hispanics and African Americans, but instead found a stronger association in census tracts with a low percentage of Hispanics and African Americans. The percentage of White people in a census tract appears to have an additive effect on the association between PM_2.5_ and LBW. While this is counterintuitive, it may be an artifact of census tracts with a high percentage of White people having lower levels of PM_2.5_ exposure compared to other racial/ethnic groups. Another possible explanation for these surprising results is that we did not have individual-level data on maternal race and only had data on the percentage of each race/ethnicity group in a census tract. Aggregated population estimates might obscure the true effect of race/ethnicity at the individual level. Lastly, there might also be residual confounding within our model, by factors such as maternal age, which increases the odds of LBW [3], or because of the known association between poverty and in what neighborhoods people from different racial and ethnic groups live. For percentage of Hispanic, White and African Americans, we cannot conclusively say that there is effect modification due to the overlap of confidence intervals. However, because the confidence intervals for below the median do not contain the point estimate for above the median (and vice versa) it suggests that there may be an effect for those three groups.

The biological mechanism for the effect of PM_2.5_ on LBW is not fully understood, but several theories have been proposed. One theory is that maternal exposure to fine particulate matter can contribute to oxidative stress, which can damage DNA [19]. In addition, maternal exposure to PM_2.5_ may lead to pulmonary and placental inflammation [20]. The effects of oxidative stress and pulmonary and placental inflammation may lead to impaired transplacental oxygen and nutrient exchange in the placenta, resulting in insufficient nutrient and oxygen delivery during gestation for optimal fetal growth [19]. Further research needs to be conducted to elucidate the biological pathways by which particulate matter affects the fetus.

One strength of this study was that it is representative of the entire state of California, with data for almost all (97%) census tracts in California being used. We also used census tracts as the unit of analysis, which is a smaller unit of analysis than other ecological studies conducted at the zip code or county level. In addition, we leveraged a unique, publicly available dataset that compiled disparate data into one platform. Lastly, this study assessed for effect modification by both socioeconomic status and race/ethnicity.

This study had several limitations. Data on both PM_2.5_ and LBW were aggregated over space and time. Because data were collected at the census tract level, the results from this study cannot be extrapolated to the individual level. PM_2.5_ was measured as the mean concentration over three years and did not take into account variations over time. Although some of the data on percent LBW were collected before the data on PM_2.5_, prior studies have shown the spatial distribution of air pollution typically does not change significantly over periods of 10 years [21]. In addition, CalEnviroScreen contained data for the total mass of PM_2.5_, with no information on the specific chemical components of PM_2.5_, which might have differential effects on LBW.

Like many other air pollution studies, this study faced possible exposure misclassification due to measurement error for PM_2.5_. The data included only measurements for outdoor air pollution and did not include any information regarding air pollution indoors, where pregnant women spend most of their time. Ambient PM_2.5_ was used as a proxy for maternal exposure. Percentage of LBW in a census tract was found by geocoding births to maternal residential addresses, thus we could not take into account residential mobility. Exposure misclassification would likely result in an underestimate of the effect of PM_2.5_ on low birth weight.

Because this was an ecologic study, future research should use individual-level data to further examine whether income and race/ethnicity modify the relationship between maternal exposure to PM_2.5_ and LBW, and should assess whether interventions to decrease PM_2.5_ exposures result in decreases in LBW.

## 5. Conclusions

This study shows that exposure to PM_2.5_ is associated with an increase in the percentage of low birth weight (LBW) infants in a census tract. Although the apparent effect of PM_2.5_ on the percentage of LBW infants was small, air pollution is a ubiquitous exposure, so the population attributable risk is more substantial than what it might appear at first glance. These study results add to the increasingly robust evidence that PM_2.5_ can have adverse effects on birth outcomes and that low-income communities are more vulnerable. Improving air quality, especially in low-income communities, could potentially improve maternal and child health outcomes.

## Figures and Tables

**Figure 1 ijerph-19-13554-f001:**
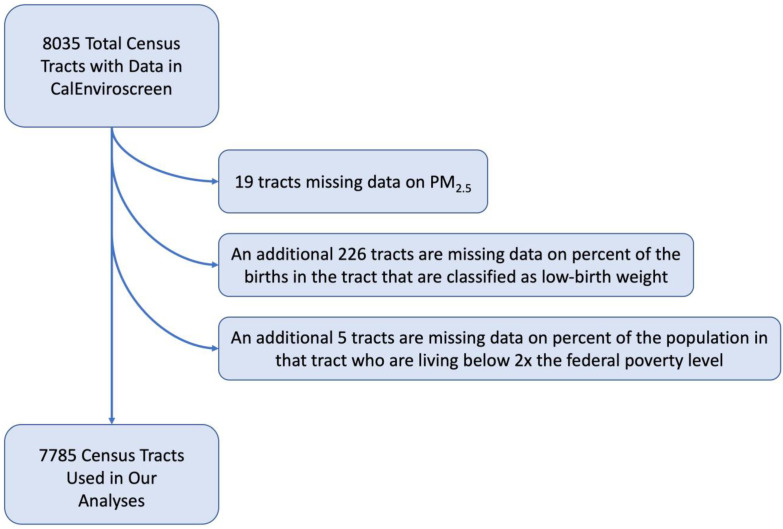
Flowchart demonstrating the selection of census tracts used in the analyses.

**Figure 2 ijerph-19-13554-f002:**
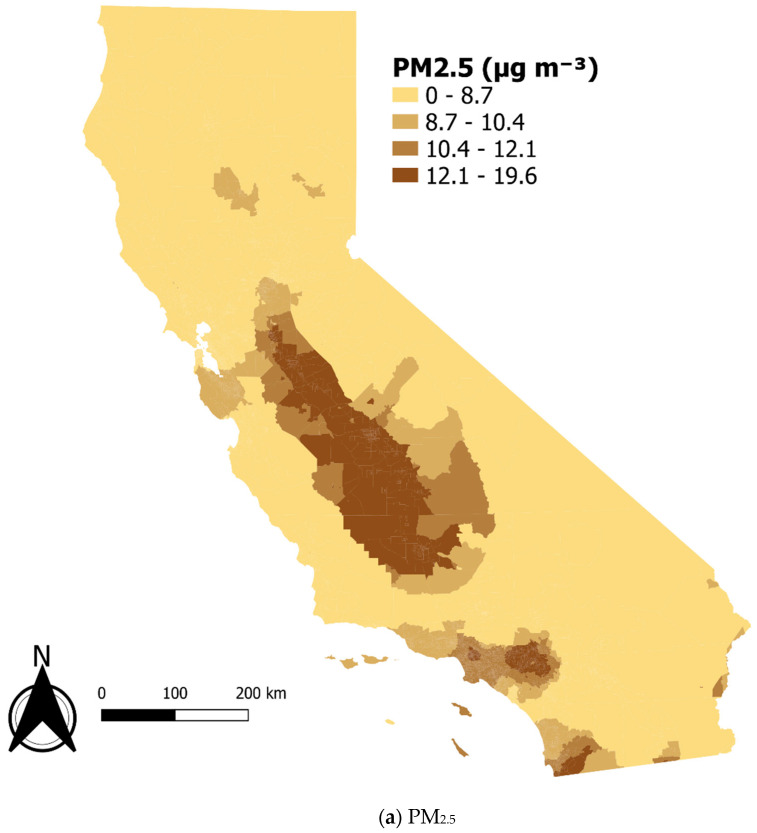
Choropleth maps of California using data from CalEnviroScreen: (**a**) PM_2.5_. (**b**) Percentage of low birth weight infants. (**c**) Percentage of households in poverty.

**Figure 3 ijerph-19-13554-f003:**
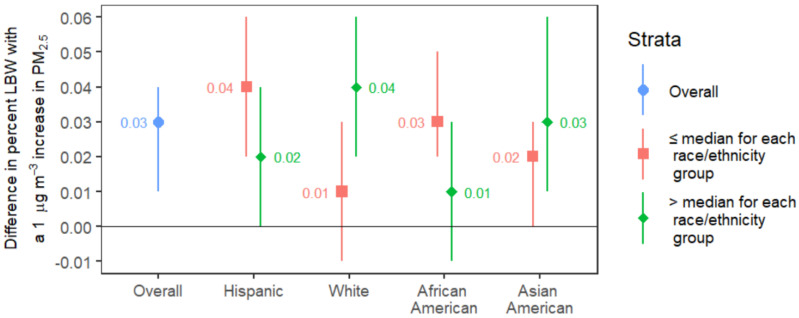
Percent change in low birth weight per 1 µg m^−3^ change in PM_2.5_ stratified by median race/ethnicity in California census tracts. The central points are the estimates of percent LBW associated with a 1 µg m^−3^ difference in PM_2.5_, and the whiskers demonstrate the 95% confidence intervals for these estimates. All models adjusted for the other race/ethnicity groups, poverty and toxic releases from facilities. Median % Hispanic: 31.4%, Median % White: 37.1%, Median % African American: 2.6%, Median % Asian American: 8.2%. Numeric results are in Appendix A.

**Table 1 ijerph-19-13554-t001:** Characteristics of 7785 census tracts in CalEnviroScreen 3.0 or 4.0. This table presents the distribution of these values from census tracts across California.

Census Tract Characteristic	25th Percentile	Median	Mean	75th Percentile	Standard Deviation
Total number	3425	4454	4708	5678	1916.9
PM_2.5_ (µg m^−3^) ^1^	8.7	10.4	10.4	12.1	2.6
Toxic Releases from Facilities (toxicity weighted lb y^−1^) ^2^	103.2	489.9	3197.5	3523.2	12,604.7
Low Birth Weight (%) ^3^	3.9	4.9	5	6	1.6
*Socioeconomic Status*					
Poverty (%) ^4^	19.2	33.5	36.4	51.6	20.3
Unemployment (%) ^5^	6.6	9.3	10.2	12.8	5
Housing Burden (%) ^6^	12.8	18	19.3	24.5	8.7
Educational Attainment (%) ^7^	6.4	14.2	19.2	28.8	16
Linguistic Isolation (%) ^8^	3.1	7.5	10.5	15	10
*Race/Ethnicity* ^3^					
Hispanic (%)	15.8	31.4	38.4	58.6	26.5
White (%)	14.4	37.1	38.4	60.6	25.9
African American (%)	0.8	2.6	5.6	6.6	8.7
Asian American (%)	3.1	8.2	13.7	18.1	15.5
Other (%) ^9^	1.8	3.4	3.9	5.5	2.9

^1^ Particulate matter less than 2.5 microns in diameter (PM_2.5_) measured as the annual mean concentration over 3 years (2012-2014) in micrograms per cubic meter at each census tract. ^2^ Pounds per year averaged over 2011–2013 at each census tract. ^3^ From CalEnviroScreen 4.0 within each census tract. ^4^ Five-year estimate from the years 2011 to 2015 of the percentage of people living below twice the federal poverty level within each census tract. ^5^ Five-year estimate from the years 2011 to 2015 of the percentage of the population over age 16 that is unemployed within each census tract. ^6^ Five-year estimate from the years 2009 to 2013 of the percentage of households that are low income and severely burdened by housing costs within each census tract. ^7^ Five-year estimate of the percentage of the population over age 25 with less than a high school education within each census tract. ^8^ Five-year estimate of the percentage of limited English-speaking households within each census tract. ^9^ Combines percentage of Native American, Pacific Islander, and Other/Multiple within each census tract.

**Table 2 ijerph-19-13554-t002:** Percent change in low birth weight per 1 µg m^−3^ change in PM_2.5_ stratified by percent poverty in California census tracts.

	Overall ^2^ (*n* = 7785)	<50% Poverty ^3^ (*n* = 5682)	≥50% Poverty ^3^ (*n* = 2103)
	β (95% CI)	β (95% CI)	β (95% CI)
PM_2.5_ ^1^	0.03 (0.01, 0.04)	0.02 (0.00, 0.03)	0.06 (0.03, 0.08)

^1^ Particulate matter less than 2.5 microns in diameter. ^2^ Adjusted for poverty, race/ethnicity, and toxic releases from facilities. ^3^ Adjusted for race/ethnicity and toxic releases from facilities.

## Data Availability

This paper used publicly available datasets. CalEnviroScreen data is available at https://oehha.ca.gov/calenviroscreen, accessed on 13 October 2022.

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
