# Peer review of "The Association between Ambient PM_2.5_ and Low Birth Weight in California"

_ijerph, 2022, doi:10.3390/ijerph192013554_

Round 1

Reviewer 1 Report

1.In study sample,It is best to have a data cleaning flow chart.

2.In Table 1, Low Birth Weight (%) is supposed to be a relative number-rate, how can quartiles be calculated? The same question is true for the quartiles of socioeconomic status and race calculated in the table.

3.Figure 1 in 7 maps show little meaning! And it takes up too much space.

Author Response

Please see the attachment, thanks!

Reviewer 2 Report

Review of “The Association Between Ambient PM2.5 and Low Birth Weight in California”

Thank you for the opportunity to review this paper. It is very well done, and can be published with only minor edits, suggested below.

Abstract

It seems odd to say that few studies control for race and poverty, since this seems to be widely recognized as an important confounder. Rewording might be in order.

The abstract should make it clearer that this study was limited to outdoor PM, not indoor, although this is mentioned in the discussion section.

This seems to be a surprisingly small effect, which is noted by the authors.

Specific comments

Line 102. The time element is missing here. It’s ug/m3 over a quarter? Or a year? Something else?

Line 105. Why are the years different for LBW estimates and PM2.5 estimates? Should they not be the same?

Line 117. Perhaps the authors can explain why twice the federal poverty level was selected instead of just using the poverty level itself? Or even low-income versus very low-income? Would the effect have been larger if a lower income level had been used?

Line 189. It might help the reader to state that this was 33% lower for higher income populations (although this sentence has many numbers, so perhaps a summary sentence is in order?)

Figure 2. Perhaps the whiskers should be stated as 95% CI? Also, what does “change” refer to exactly? What is the reference group exactly?

Line 268. Perhaps the co-linearity between race and income should be mentioned here.

Line 288. Stating the obvious, future research should include the effects of interventions.

Author Response

Please see the attachment, thanks!
